# Molecular Docking Studies of Coumarins Isolated from Extracts and Essential Oils of *Zosima absinthifolia* Link as Potential Inhibitors for Alzheimer’s Disease

**DOI:** 10.3390/molecules24040722

**Published:** 2019-02-17

**Authors:** Songul Karakaya, Mehmet Koca, Serdar Volkan Yılmaz, Kadir Yıldırım, Nur Münevver Pınar, Betül Demirci, Marian Brestic, Oksana Sytar

**Affiliations:** 1Department of Pharmacognosy, Faculty of Pharmacy, Ataturk University, 25240 Erzurum, Turkey; ecz-songul@hotmail.com (S.K.); svyilmaz@yandex.com (S.V.Y.); kdryldrm25@gmail.com (K.Y.); 2Department of Pharmaceutical Chemistry, Faculty of Pharmacy, Ataturk University, 25240 Erzurum, Turkey; kocamehmet@atauni.edu.tr; 3Department of Biology, Faculty of Science, Ankara University, 06560 Ankara, Turkey; Nur.M.Pinar@science.ankara.edu.tr; 4Department of Pharmacognosy, Faculty of Pharmacy, Anadolu University, 26210 Eskisehir, Turkey; betuldemirci@gmail.com; 5Department of Plant Physiology, Slovak Agricultural University in Nitra, 94976 Nitra, Slovak; marian.brestic@uniag.sk; 6Department of Plant Biology, Educational and Scientific Center “Institute of Biology and Medicine”, Kiev National University of Taras Shevchenko, Hlushkova Avenue, 2, 03127 Kyiv, Ukraine

**Keywords:** Apiaceae, antioxidant, anticholinesterase, essential oil, secretory canals, *Zosima absinthifolia*

## Abstract

Coumarins and essential oils are the major components of the Apiaceae family and the *Zosima* genus. The present study reports anticholinesterase and antioxidant activities of extracts and essential oils from aerial parts, roots, flowers, fruits and coumarins—bergapten (**1**); imperatorin (**2**), pimpinellin (**3**) and umbelliferone (**4**)—isolated of the roots from *Zosima absinthifolia*. The investigation by light and scanning electron microscopy of the structures of secretory canals found different chemical compositions in the various types of secretory canals which present in the aerial parts, fruits and flowers. The canals, present in the aerial parts, are characterized by terpene hydrocarbons, while the secretory canals of roots, flowers and fruits include esters. Novel data of a comparative study on essential oils constituents of aerial parts, roots, flowers and fruits of *Z. absinthfolia* has been presented. The roots and fruits extract showed a high content of total phenolics and antioxidant activity. The GC-FID and GC-MS analysis revealed that the main components of the aerial parts, roots, flowers and fruits extracts were octanol (8.8%), octyl octanoate (7.6%), octyl acetate (7.3%); *trans*-pinocarvyl acetate (26.7%), β-pinene (8.9%); octyl acetate (19.9%), *trans-p*-menth-2-en-1-ol (4.6%); octyl acetate (81.6%), and (*Z*)-4-octenyl acetate (5.1%). The dichloromethane fraction of fruit and flower essential oil was characterized by the highest phenolics level and antioxidant activity. The dichloromethane fraction of fruit had the best inhibition against butyrylcholinesterase enzyme (82.27 ± 1.97%) which was higher then acetylcholinesterase inhibition (61.09 ± 4.46%) of umbelliferone. This study shows that the flowers and fruit of *Z. absinthifolia* can be a new potential resource of natural antioxidant and anticholinesterase compounds.

## 1. Introduction

Alzheimer’s disease (AD) is a degenerative brain disease and the most widespread reason for dementia. The characteristical symptoms of dementia are troubles with memory, language, problem-solution and other cognitive abilities that influence a person’s ability to make daily activities. These troubles happen because nerve cells in parts of the brain involved in cognitive function have been ruined. In AD, neurons in other parts of the brain are finally damaged or destroyed as well, including those that permit a person to perform basic bodily functions such as walking and swallowing. People in the final stages of the disease are bed-bound and require around-the-clock care. AD is eventually fatal [1]. Many factors such as age are risk factors in AD. Due to the ageing population, it is expected that AD will become a serious socio-economic challenge globally in the coming years [2]. With reference to the World Health Organization data, AD, which affects about 47 million people worldwide, is the most pervasive form of dementia (60–80% of all cases) [3] with a proximate worldwide cost of US$818 billion [4]. Oxidative stress, occurring through a damage to neurons or metal accumulation has been related to the pathogenesis of AD. Drugs endowed with anticholinesterase and antioxidant capacity could thus be useful for the prevention/treatment of AD [5].

To overcome the limitations of current therapeutics for AD, extensive research is under way to identify drugs that are both effective and free of undesired side effects. In this context naturally occurring dietary polyphenolic phytochemicals have received remarkable attention as alternative options for AD treatment.

In particular, curcumin, resveratrol, and green tea catechins have been identified to have the potential to prevent AD owing to their anti-amyloidogenic, anti-oxidative, and anti-inflammatory characteristics. These polyphenolic phytochemicals also activate adaptive cellular stress responses, called ‘neurohormesis’, and supress illness processes [6]. Antioxidants may trap reactive oxygen species (ROS) and break inflammatory pathways. The utilization of antioxidants is useful to delay AD progress [7]. A particular and important preventive action against AD with hop iso-α-acids, which are reponsible for the bitterness in beer, was discussed lately. Besides, proof has appeared for anti-carcinogenic action from hops’prenylflavonoids, as well as from phenolic components extracted from both malt and hops [8]. Numerous investigations were performed on the biological activities of plants which are utilised traditionally as memory enhancers and acetylcholinesterase inhibitors [9,10]. Representatives of the family Apiaceae demonstrate acetylcholinesterase inhibitory activity [10,11]. Natural compounds of phenolic nature have shown a substantial role in the inhibition of acetylcholinesterase enzyme (AChE) [12,13]. Phenolic compounds of medicinal plants and dietary plants are present as coumarins, curcuminoids, flavonoids, lignans, phenolic acids, tannins, stilbenes, quinones, and others. The varied bioactivities of phenolic compounds are responsibe for their AChE inhibition capacities [12,14].

At the same time, essential oils are verified to provide varied pharmacological effects, like antiflatulent, antiviral, antispasmodic, anticarcinogenic, and hepatoprotective effects, etc. Essential oils have been reported to be natural antioxidants and proffered principally as potential substitutes of synthetic antioxidants used the in food conservation sectors. Further, biologically active natural compounds can be used in the pharmaceutical industry to check human sicknesses of microbial origin and treat lipid peroxidative damage, which is observed in certain pathological disorders, such as AD, carcinogenesis, ischemia-reperfusion injury, coronary atherosclerosis, and ageing processes [14]. Antioxidants comprise most of the active ingredients of the $80 billion anti-ageing product market, which is growing at >10% yearly growth rate. Olive polyphenols—whose hydroxytyrosol and verbascoside compounds share the highest grade of antioxidant activity ever reported for any natural compound—can be effectively utilized in for health, appearance enhancement, and fitness purposes as well as in the anti-ageing products market [15].

Representatives of the families Apiaceae and Lamiaceae are characterized by high phenolics content [16] and were demonstrated to have positive effects on the central nervous system [4]. 

*Z. absinthifolia* Link is the only member of the *Zosima* genus that grows in Turkey, where it is commonly known as ‘ayı eli’ or ‘peynir otu’. The aerial parts of the plant are used up as a vegetable and added to a traditional cheese in East Anatolia. In folk medicine, fruits of the plant have digestive and sedative effects with anti-inflammatory properties. Moreover, the aerial parts cure dyspepsia, stomach gas, cough and intestinal disorders [17]. Coumarins, such as deltoin and columbianadin, have also been isolated from *Z. absinthifolia* [18]. It has been reported that *Z. absinthifolia* has biological activities such as cytotoxic, antioxidant, antibacterial, anti-inflammatory [19,20] and antimycobacterial effects [21]. Previous phytochemical studies have demonstrated that *Z. absinthifolia* contains alkaloids and coumarins such as deltoin, imperatorin, zosimine, pimpinellin, bergapten, isobergapten, sphondin isopimpinellin, and umbelliferone [17].

The presented research studied the cholinesterase inhibitory, antioxidant activity, and phenolics content of the methanol, hexane, dichloromethane, ethyl acetate, butanol and aqueous extracts and essential oils of aerial parts, roots, flowers and fruits of *Z. absinthifolia*. The AChE and BuChE inhibitory activities of the coumarins bergapten (**1**), imperatorin (**2**), pimpinellin (**3**) and umbelliferone (**4**) isolated from the roots of *Z. absinthifolia* were also assessed through molecular docking studies with parallel investigation of the structures of the plant’s secretory canals.

## 2. Results

The CH_3_OH extracts of aerial parts, roots, flowers and fruits of *Zosima absinthifolia* were fractionated with the use of different solvents (*n*-hexane, dichloromethane, ethyl acetate and butanol), to give the respective fractions and the individual coumarins bergapten (**1**), imperatorin (**2**), pimpinellin (**3**) and umbelliferone (**4**) isolated from roots which were assayed for antioxidant, acetylcholinesterase and butyrylcholinesterase inhibitory activities. Also, the AChE and BuChE inhibitory activities of the compounds were determined via molecular docking. 

The active dichloromethane fraction of root was subjected to column chromatography over silica gel and Sephadex LH-20. As the result, four known coumarins namely, bergapten (**1**) [17], imperatorin (**2**) [17], pimpinellin (**3**) [22] and umbelliferone (**4**) [17] (Figure 1) were isolated and identified in several places before it says these were isolated from the roots—explain. 

The extracts, fractions and essential oils of aerial parts, roots, flowers and fruits were studied regarding their antioxidant capacity potential. The findings of content of total phenolics from the samples are presented in Figure 2B. The highest level of total phenolics was seen in root and fruit (59.81 and 52.34 mg GAE g^−1^ DW, respectively) while the least content of phenolics was seen in the aerial parts of the plant (34.07 mg GAE g^−1^ DW). DPPH analysis results showed the presence of antioxidant activity in the range from 61.92–69.2% with the highest seen in the fruits compared to the extracts of the aerial parts of the plant (Figure 2A). The antioxidant activity results of extract from different *Z. absinthifolia* parts were quite high compared with the standards propyl gallate, chlorogenic acid, and rutin (Table 1). 

Table 1 shows the TBA assay results of the specimens reported as IC_50_ values (μg mL^−1^). The highest antioxidant potential in the TBA assay was seen in the fruit CH_2_Cl_2_ fraction and flower essential oil (IC_50_ = 48.98 and 97.11 μg/mL, respectively). Among the isolated compounds pimpinellin and bergapten had strong antioxidant effects, with IC_50_ values of 49.23 and 56.99 μg/mL. Many of samples indicated considerable antioxidant activity on liposomes but not comparable to rutin or chlorogenic acid. The correlation coefficient between antioxidant capacity and content of total phenolics is remarkable (0.96). 

The anticholinesterase activity of the samples was assessed by means of the Ellman colourimetric method [23], with a few changes and using commercially available donepezil as reference [24]. The in vitro anti-acetylcholinesterase activities of the specimens at 20 µg/mL are presented in Table 2. The MeOH, hexane, CH_2_Cl_2_, EtOAc and BuOH extracts and fractions of essential oils from all plant parts demonstrated significative inhibitory activities towards butyrylcholinesterase. The fruit CH_2_Cl_2_ fraction and flower essential oil indicated considerable inhibition against BuChE (82.27 ± 1.97 and 78.65 ± 2.66%, respectively). The CH_2_Cl_2_ fractions of root and fruit also showed considerable inhibition against AChE (29.15 ± 2.45 and 31.46 ± 2.78%, respectively). Among the isolated compounds umbelliferone indicated strong inhibition against AChE (61.09 ± 4.46%) and pimpinellin had strong inhibition against BuChE with a 66.55 ± 2.61% value. On the other hand, none of the aqueous residues had activity against AChE, while only aerial part essential oil had no activity against this enzyme. The BuOH fraction of fruit and aqueous residue fraction of aerial parts and flowers displayed no butyrylcholinesterase inhibition activity. Amongst the essential oils the fruit (83.01%) and root (81.32%) ones indicated considerable inhibition towards BuChE. 

The most active compounds (umbelliferone against AChE and pimpinellin against BuChE) were docked at the binding sites of 1-EVE and 1-P0I. The molecular interactions of the compounds possibly accounting for the inhibition are shown in Figure 3 and Figure 4. Umbelliferone exhibits a good docking score for 1-EVE (−7.46 kcal/mol) when compared to the standard donepezil. Umbelliferone has three π-π stacking interactions (4.25 Å, 3.89 Å and 4.70 Å) with PHE330 and TRP84. In addition, hydrophobic interactions were formed between the molecule and TRP84, PHE330, PHE331, TYR121, TYR334. The polar interaction was realized by HIS 440. On the other hand the docking score of pimpinellin was −5.78 kcal/mol compared to the standard donepezil. Pimpinellin has two π-π stacking interactions (5.09 Å and 5.10 Å) with the phenyl ring of PHE 329. In addition, hydrophobic interactions were formed between the molecule and the PHE329, PRO285, LEU286, VAL288, TRP231, PHE398, ALA199 residues. The polar interactions were realized by SER287, GLN119, SER198.

Essential oil % yields of the various parts and the colours of these essential oils are presented in Table 3. The colours of essential oils from different parts of *Z. absinthfolia* varied. The flowers and fruits essential oils of *Z. absinthfolia* were yellow while the aerial part and roots gave light yellow and white coloured oils, respectively. 

In general, the yield of the root oil was low compared to the aerial part, fruit and flower ones. The best yield results were obtained for fruit (Table 3). A total of thirty-three compounds totaling 94.7% of the oil were identified in the essential oil of aerial parts of *Z. absinthfolia.* Octanol, octyl octanoate and octyl acetate were the primary components, amounting to 8.8%, 7.6% and 7.3%, respectively. The analysis of the roots of *Z. absinthfolia* resulted in the identification of forty-four compounds totaling 81.6% of the oil. *trans*-Pinocarvyl acetate at 26.7% was the most abundant compound in the essential oil, followed by β-pinene (8.9%). Eighty-three compounds were characterized in the oil of the flowers of *Z. absinthfolia*, accounting for 82.5% of the oil. The primary constituents were identified as octyl acetate (19.9%), and *trans-p*-menth-2-en-1-ol (4.6%). The analysis on the fruits of *Z. absinthfolia* resulted in the determination of fifty-two essential compounds totaling 99.2%. Octyl acetate at 81.6% was the most abundant compound in the essential oil followed by (Z)-4-octenyl acetate (5.1%). The compositions of essential oils are presented in Table 4. 

The determined compounds were categorized into two main classes on the basis of their chemical structures: isoprenoids (oxygenated monoterpenes, terpene hydrocarbons) and nonisoprenoids variously functionalized (alkanes, aldehydes, lactones, ketones, alcohols, furans, fatty acids and esters). Terpene hydrocarbons, esters, fatty acids-esters, and alcohols were the dominant groups of compounds in the essential oils (Table 5).

The micrographs of the peduncles, rays, pedicels, and fruits of *Z. absinthifolia* were obtained from alcohol samples utilizing light microcopy (Figure 5, Figure 6, Figure 7 and Figure 8) and from the dried samples through Scanning Electron Microscopy (SEM, Jeol JSM 6490LV) (Figure 9a–k). The number of secretory canals in the centre was less than in the cortex at the peduncle. At the ray and pedicel secretory canals were only found in the cortex and the number of canals are higher. The secretory canals in fruit were very large and wide. 

Secretory structures of stem, leaf, flower and fruit samples of *Z. absinthifolia* were studied in detail using light and scanning electron microscopy. The plant has secretory trichomes in the leaf, stem, pedicel and fruit. There are two types of glandular trichomes; capitate trichomes and sessile peltate trichomes. The capitate trichomes were identified on the leaf, pedicel and stem, peltate trichomes on pedicel and fruit. The capitate trichomes are composed of multi basal cells, a long stalk cell with the unicellular secretory head. Peltate trichomes exhibit a flattened head in the pedicel or a granular head in fruit formed by several cells arranged in a circle (Figure 9). Extrafloral nectaries are found on the pedicel. The secretory ducts show a lumen surrounded by a layer of specialized cells in fruit. Excretion secretory system organs including crystals are observed in the fruit. 

## 3. Discussion

Coumarins are compounds naturally present in a great number of plants. Coumarin and its derivatives are prevalent in Nature. Coumarins are benzopyrones, which are compounds comprised of benzene rings linked to a pyrone moiety. Human dietary exposure to benzopyrones is quite considerable, as these compounds occur in fruits, vegetables, seeds, nuts, and higher plants. It has been determined that the mean Western diet includes ~1 g/day of mixed benzopyrones [25]. Coumarins have various biological activities such as anticancer, anticoagulant, anti-inflammatory, antitubercular, antihyperglycemic, antiadipogenic, antifungal, antibacterial, anticonvulsant, antihypertensive, antiviral, antioxidant, neuroprotective and antidiabetic effects [26].

In the current research, umbelliferone and pimpinellin were isolated from *Zosima absinthfolia*, and indicated activity against AChE and BuChE. We assumed that the inhibitory activity of umbelliferone was primary due to the hydroxyl group at the C-7 position. Also, we assumed that the inhibitory activity of pimpinellin was primary due to the methoxy groups at the C-5 and C-6 positions. Umbelliferone and pimpinellin presented the best activity at 20 µg/mL, while bergapten indicated weak inhibitory activity. 

The roots fractions have been characterized by the significant higher content of total phenolics than aerial part fractions. Previously, it was observed that methanol fruit extract of *Z. absinthfolia* showed high antioxidant activities and a greater content of total phenols in comparison to the hexane and dichloromethane extracts of the plant [27]. A significant correlation was observed between antioxidant capacity and the total phenolics content in previous investigations [28,29] as well. The antioxidant capacity of *Z. absinthfolia* plant extracts was studied. It was found that peroxidation inhibition of MeOH extract of *Z. absinthfolia* was 143.5 RC_50_ [27]. However, we found that, fruit CH_2_Cl_2_ fraction showed higher antioxidant activity in comparison with the MeOH extract.

In a previous study, the major components of essential oils from different parts of *Z. absinthfolia* were studied and it was shown that the main volatile compounds were lavandulyl acetate (23.9%, an ester of the irregular monoterpenol), bornyl acetate (12.0%), octyl octanoate (11.7%), lavandulol (5.0%), octyl hexanoate (4.2%) and lavandulyl octanoate (3.1%) [30]. Another study reported that the major components of oil from the aerial part of Z. *absinthfolia* were octyl acetate (32.50%), octanol (20.60%) and α-pinene (10.90%) [31]. Octyl acetate (87.48%), octyl octanoate (5.03%) and 1-octanol (2.37%) were found the major components of fruit essential oil [19]. Octyl acetate (38.4%) and octyl hexanoate (31.9%) were detected as major components of fruit essential oil of *Z. absinthfolia* [32]. Our study is the first comparative study on essential oils constituents of aerial parts, roots, flowers and fruits of *Z. absinthfolia* and their anticholinesterase and antioxidant activities.

So far, no data are available about the presence of phenolic compounds in the essential oil from *Zosima* genus, especially such as α-pinene and octyl acetate which in previous studies showed high antioxidant capacity [33,34]. The high abundance of octyl acetate and possible antioxidative and anticancer effects were estimated in essential oils from the leaves of species of *Pittosporum* (Pittosporaceae) [35] and in essential oils from Ethiopian herbs *Boswellia carterii* and *Commiphora pyracanthoides* Engler [36].

The chemical composition of the different types of secretory canals present in the aerial parts, fruits and flowers were different. The canals present in the aerial parts are characterized by terpene hydrocarbons, while the secretory canals of roots, flowers and fruits include esters. Only canals of fruits contain lactones. Besides, canals of flowers and fruits contain alkanes. We can comment that the number of secretory channels, their location in the organs and their different shapes (broad, big, little, oval etc) could be related to the different chemical compositions of aerial part, root, flower and fruits. The chemical class distribution of the samples is presented in Table 5.

Members of the Apiaceae family are characterized by a specific type of essential oil secretory structure known as secretory canals. Their shapes and numbers can vary between species, within species or even in individual plants. They have large amounts of metabolic products in the area between their secretory canals. Particularly they generate and store essential oils in plants [37,38].

AD is a neurodegenerative disease induced by oxidative stress with a further cholinergic deprivation in the brain. Expressly, a decrease in the amount of acetylcholine delivered from cholinergic synapses has been identified as a cause. One cure methodology involves augmenting or protecting the ratio of acetylcholine via inhibiting acetylcholinesterase [39]. This research indicated that the CH_2_Cl_2_ fraction of fruit from *Z. absinthifolia* has AChE and BuChE inhibitory activity along with high antioxidant capacity. The use of antioxidants may be useful to treat AD. To the knowledge of the authors, this is the first study on the anticholinesterase activity of extracts from *Z. absinthifolia.*


Essential oil components represent a diverse family of low molecular weight organic compounds with remarkable biological activity. In accordance with their chemical structure, these active compounds can be divided into four major groups: terpenoids, phenylpropenes, terpenes, and “others”. Besides, they might include diversified functional groups in accordance with which they can be categorised as hydrocarbons (monoterpenes, sesquiterpenes, and aliphatic hydrocarbons); oxygenated compounds (monoterpene and sesquiterpene alcohols, esters, ketones, aldehydes, and other oxygenated compounds); and sulfur and/or nitrogen sulfur including compounds (sulfides, nitriles, thioesters, isothiocyanates, and others). Components that act as cholinesterase inhibitors still represent the only pharmacological treatment of AD. Many in vitro investigations have demonstrated that some compounds present in essential oils such as α-pinene, α- and β-asarone, δ-3-carene, carvacrol, 1,8-cineole, thymohydroquinone, anethole, etc have certain cholinesterase inhibitory activity [40].

## 4. Materials and Methods

### 4.1. Plant Specimens

*Zosima absinthifolia* samples were gathered at the flowering and fruity period from Erzurum in the Palandöken Mountains in 2017 and 2018, and the verified by Prof. Dr Hayri Duman. Voucher specimens were stored at the Herbarium of the Atatürk University Faculty of Pharmacy (AUEF 1275 and AUEF 1283).

### 4.2. Extraction and Isolation

Aerial parts (150 g), roots (150 g), flowers (150 g) and fruits (500 g) were comminuted and macerated with methanol (3 times × 8 h) in a water-bath not exceeding 40 °C (3 × 150 mL) while mixing at 300 rpm with the use of a mechanical mixer. Conjoined aerial parts, roots, flowers and fruits extracts were filtered and concentrated up to dryness using a rotating evaporator (Heidolph VV2000, Schwabach, Germany). After that the residue was dissolved in methanol:water (1:9) and subjected to three further fractionation steps with 150 mL of *n*-hexane, dichloromethane, ethyl acetate and *n*-butanol, respectively. The weights of the comminuted parts of *Zosima absinthifolia* and extracts/fractions obtained are indicated in Table 6.

The extraction, and identification of purified compounds from the CH_2_Cl_2_ fruit fraction was done according to [41]. The effective CH_2_Cl_2_ fraction of fruit was first applied to a silica gel column and eluted with a gradient of hexane:EtOAc (100:0 → 0:100, *v*/*v*) and EtOAc:MeOH (100:0 → 0:100, *v*/*v*), and three fractions (Fr. A–C) were acquired. Repetitive silica gel column chromatography with hexane:EtOAc (85:15 and 80:20) solvent systems on Fr. A gave compound **1**. Fr. B was applied to a silica gel column and eluted with hexane:EtOAc (75:25) and a Sephadex LH-20 column eluting with ethyl acetate to give compounds **2** and **3**. Elution with hexane:EtOAc (70:30) of a silica gel column of Fr. C gave compound **4**. The chemical structures of compounds **1**–**4** are presented in Figure 1.

### 4.3. Isolation of the Essential Oil, GC-FID and GC-MS Analyses

Isolation of the essential oils, GC-FID and GC-MS analyses processes were performed according to [42]. The crushed parts, essential oil % yields of the species and colours of essential oils are presented in Table 2.

### 4.4. Determination of Total Phenolics

The total phenolic content of specimens was evaluated utilising the Folin–Ciocalteu assay [43] with slight modifications [44]. The total phenolics absorbance was determined at 765 nm with the use of a Jenway UV/Vis 6405 spectrophotometer (Jenway, Chelmsford, UK). The findings are reported as gallic acid equivalents (GAE/g specimens).

### 4.5. 1,1-Diphenyl-2-picrylhydrazyl (DPPH) Radical Scavenging Capacity Assay

The previously detailed DPPH assay [45] was applied with slight alterations. Reagent stock solution (1 × 10^−3^ M) was prepared by dissolving 22 mg of DPPH in 50 mL of methanol. This solution was kept at 20 °C until used. Samples (0.02 g) were extracted in two steps: first, to the dry material in an Eppendorf tube was added 1 mL of distilled water. Specimens were heated at 95 °C during 15 min and further for 5 min centrifuged (12,000 rpm, 25 °C). The supernatant was transferred into a fresh tube. The supernatant with 1 mL of dist. water was diluted and the same heating and centrifugation procedure was repeated. The study solution (6 × 10^−5^ M) was prepared by mixing 100 mL of methanol with 6 mL of the stock solution. Then, 0.1 mL of each experimental solution was mixed in to react with 3.9 mL of the solution of DPPH, followed by vortexing for 30 s and a further reaction time of 30 min. The optical absorbance was gauged at 515 nm with the Jenway UV/Vis 6405 spectrophotometer. A sample without DPPH solution was utilized as a blank sample. The scavenging activity of DPPH was measured according to the formula below:Scavenging activity of DPPH (%) = [(A control − A sample)/Acontrol] × 100where A = absorbance at 515 nm.

### 4.6. Anti-Lipid Peroxidation Activity

The thiobarbituric acid (TBA) assay was utilised to assess the protective activity of samples on liposomes against lipid peroxidation [46]. Seven different sample concentrations (0.016–1 mg/mL) were studied in this test. Chlorogenic acid, rutin and propyl gallate were prepared as reference compounds at seven different concentrations (0.000064–1 mg/mL), and chlorogenic acid and rutin were utilised in the same concentration interval. Brain extract (0.2 mL), phosphate buffer (0.5 mL), ferric chloride (0.1 mL), ascorbic acid (0.1 mL) and the samples were mixed and incubated at 37 °C for 20 minutes. Next, 25% HCl (0.5 mL), 1% TBA (0.5 mL) and 2% BHT (0.1 mL) were added into the mixture, which was shaken and incubated at 85 °C for 30 minutes. The mixture was cooled and *n*-butanol (2.5 mL) was added. After centrifugation, the absorbance of the samples was recorded at 532 nm using the UV-1800 spectrophotometer. These tests were repeated four times. The IC_50_ values were established through linear regression analysis. A low IC_50_ value means that the antioxidant activity is high.

### 4.7. Determination of AChE and BuChE Inhibition Activities

The determination of AChE and BuChE inhibition activities of the samples were performed according to [41]. This process was repeated three times for each plate. All data were expressed as mean ± SE of three independent assays.

### 4.8. Microscopic Analysis

Materials (kept in 70% alcohol) from *Zosima absinthifolia* were assessed by light microscopy using Sartur and chloral hydrate reagents. In the light microscopy study, cross-sections of peduncles, rays, pedicels, and fruits from *Z. absinthifolia* were prepared manually. Images prepared with Sartur R [47,48] were recorded with a Zeiss 51425 camera attached to a light microscope (Zeiss 415500-1800-000). In the scanning electron microscopy (SEM) investigations, leaf, stem, pedicel and fruit parts were attached to aluminium stubs and covered with gold for 4 min in a sputter-coater. Morphological observations were done in a Jeol JSM 6490LV scanning electron microscope at the Turkish Petroleum International Company (TPAO) Research Centre SEM laboratory, Ankara.

### 4.9. Molecular Docking Studies

Umbelliferone was found to be an active compound against AChE and pimpinellin was found to be active against BuChE was found. These active compounds were docked at the binding sites of 1-EVE and 1-P0I.

### 4.10. Protein Preparation

The three-dimensional complex structures of AChE (PDB ID: 1EVE) and BuChE (PDB ID: 1P0I) were obtained from the Protein Data Bank [49,50]. The protein structures were prepared using the Protein Preparation Wizard panel tool of the Scrödinger software suite (Maestro 11.8). Firstly water molecules (>5Å radius) and other small molecules were removed from the crystal structures, hydrogen atoms were added and physiological pH was set at 7. Finally, the restrained minimization was performed with the added hydrogen atoms to OPLS3e.

### 4.11. Ligand Preparation

The ligands were prepared for docking with using the Ligand Preparation Panel in the programme. The grid files were created using the Receptor Grids Generation Panel. Finally the Glide Ligand Docking Panel was used for docking studies.

### 4.12. Statistical Analysis

All findings are stated as mean ± SE and variations between means were statistically analyzed through One-way analysis of ANOVA followed via Bonferroni’s complementary analysis, with *p* < 0.05 considered to demonstrate statistical significancy.

## 5. Conclusions

The CH_2_Cl_2_ fraction of fruit from *Zosima absinthifolia* and umbelliferone had a remarkable antioxidant and anticholinesterase activities. The tested extracts and essential oils displayed high radical scavenging capacity (RSC), which was found to be in correlation to their content of phenolic compounds. Octyl acetate was the dominant component in the essential oils. Original information has been presented regarding the total phenolics content and high presence of chlorogenic acid and flavonoid rutin in the extracts and essential oils of plant *Z. absinthifolia.* Novel data of a comparative study on the essential oil constituents of aerial parts, roots, flowers and fruits of *Z. absinthfolia* has shown different phenolic compositions which can depend from the function of secretory canals of the various plant parts. Due to the remarkable presence of compounds with high anticholinesterase activities in the plant we presume that *Z. absinthifolia* could be utilised as an herbal alternative to synthetic drugs in the prophylaxis of AD. 

## Figures and Tables

**Figure 1 molecules-24-00722-f001:**
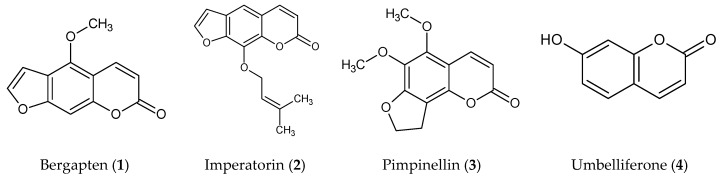
Chemical structures of compounds **1**–**4**.

**Figure 2 molecules-24-00722-f002:**
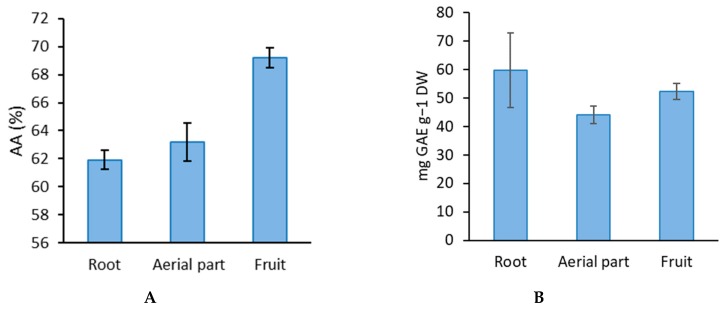
DPPH radical scavenging activity (**A**), total phenolic contents (**B**) of samples.

**Figure 3 molecules-24-00722-f003:**
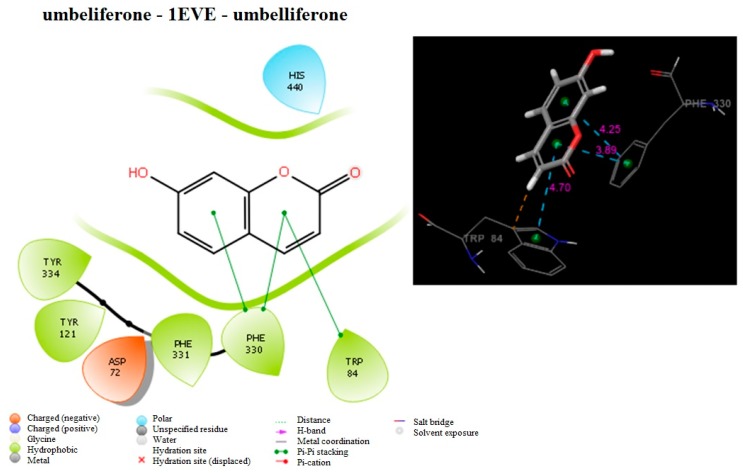
Schematic representation of main interaction of umbelliferone with AChE (1-EVE). Green color represents hydrophobic interactions, light blue represents polar interactions, blue represents positively charged residues, red represents negatively charged residues.

**Figure 4 molecules-24-00722-f004:**
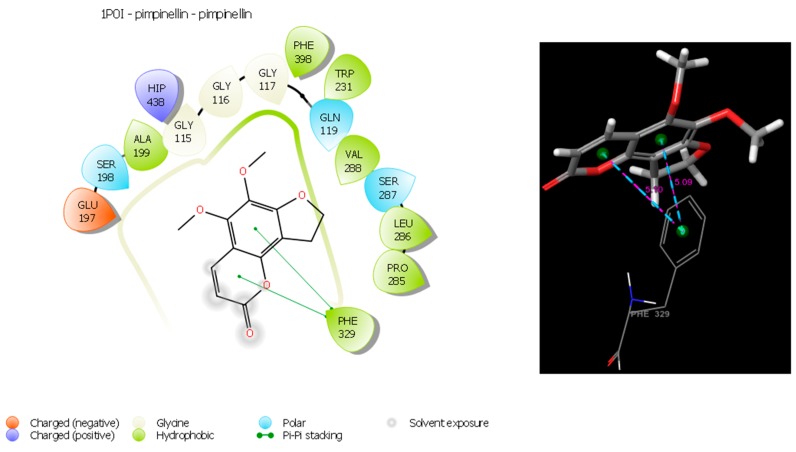
Schematic representation of the main interaction of pimpinellin with BuChE(1-P0I) Green colour represents hydrophobic interactions, light blue represents polar interactions, blue represents positively charged residues, red represents negatively charged residues.

**Figure 5 molecules-24-00722-f005:**
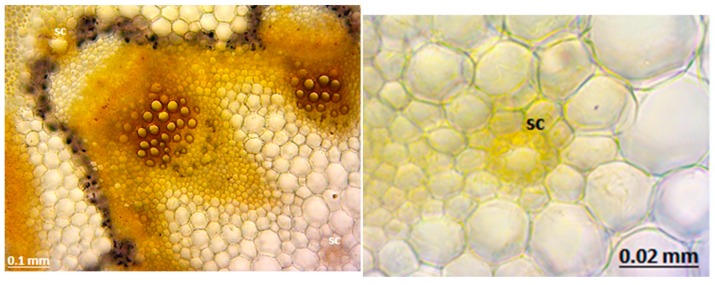
Secretory canals at the peduncle of *Zosima absinthifolia* by light microscopy.

**Figure 6 molecules-24-00722-f006:**
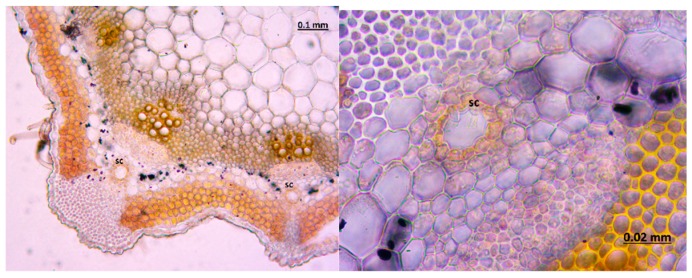
Secretory canals at the ray of *Zosima absinthifolia* by light microscopy.

**Figure 7 molecules-24-00722-f007:**
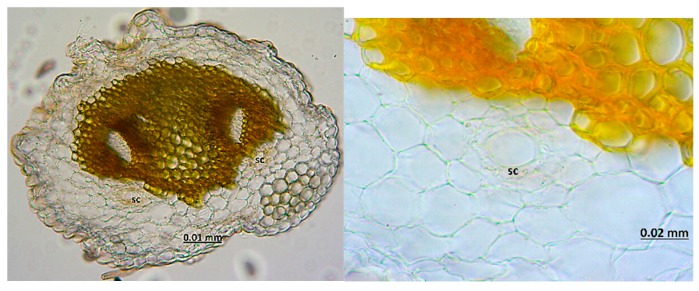
Secretory canals at the pedicel of *Zosima absinthifolia* by light microscopy.

**Figure 8 molecules-24-00722-f008:**
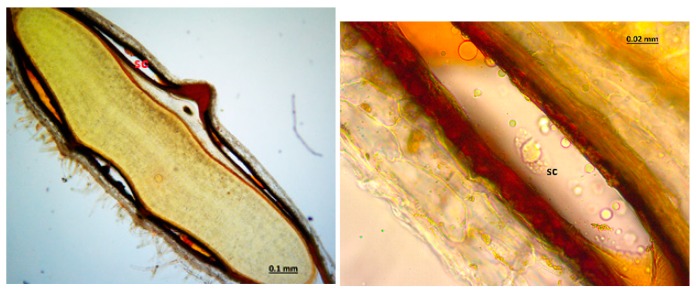
Secretory canals at the fruit of *Zosima absinthifolia* by light microscopy.

**Figure 9 molecules-24-00722-f009:**
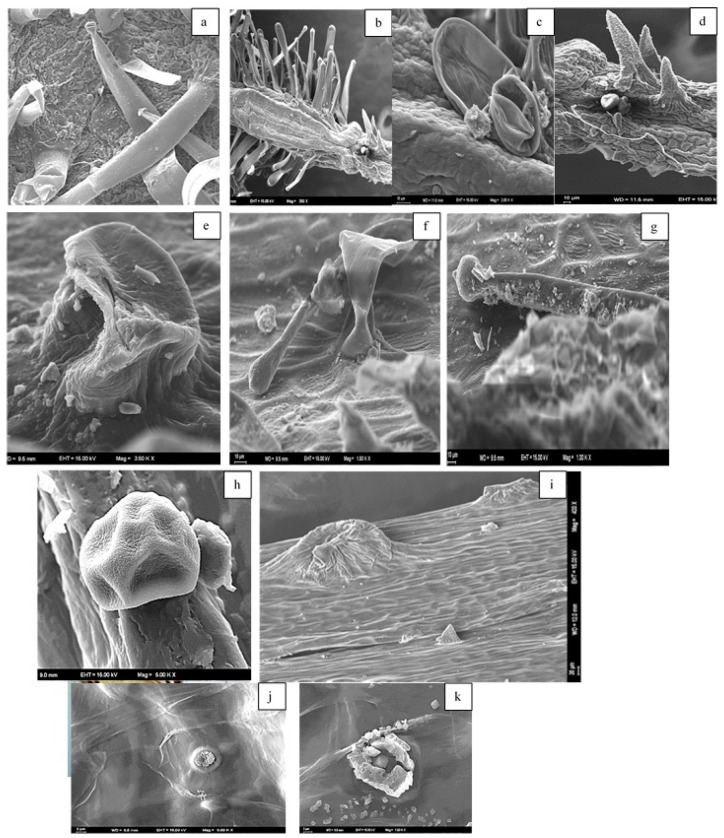
(**a**) Capitate trichomes on the leaf by SEM, (**b**–**d**) capitate trichomes on the pedicel by SEM, (**e**–**g**) capitate trichomes on the stem by SEM, (**h**–**k**) extrafloral nectaries, the secretory ducts and excretion secretory system on the fruit by SEM.

**Table 1 molecules-24-00722-t001:** Antioxidant activities of the samples from *Zosima absinthifolia* in thiobarbituric acid (TBA) test.

Tested Samples	IC_50_ Values (µg/mL) ± SD *			
	Aerial Part	Root	Flower	Fruit
MeOH	204.16 ± 2.16	234.21 ± 4.26	199.43 ± 2.46	116.25 ± 3.51
Hexane	266.67 ± 2.97	198.15 ± 1.78	159.54 ± 1.48	500>
CH_2_Cl_2_	169.21 ± 4.22	500>	301.53 ± 3.01	48.98 ± 2.45
EtOAc	255.21 ± 3.43	217.99 ± 4.35	478.90 ± 1.78	196.25 ± 2.66
BuOH	500>	367.60 ± 3.21	298.33 ± 3.55	487.45 ± 3.12
Aqueous residue	500>	500>	500>	500>
Essential oils	225.17 ± 3.29	390.36 ± 1.56	97.11 ± 2.25	389.67 ± 2.86
Bergapten	56.99 ± 3.87			
Imperatorin	79.23 ± 3.48			
Pimpinellin	49.23 ± 2.19			
Umbelliferone	79.53 ± 3.98			
Chlorogenic acid	12.98 ± 4.89			
Propyl gallate	3.44 ± 2.05			
Rutin	9.65 ±3.09			

* Standard deviation.

**Table 2 molecules-24-00722-t002:** In vitro AChE and BuChE inhibitory activities of samples from *Zosima absinthifolia* at 20 µg/mL.

Samples	Enzymes	Percentile of inhibition ± S.E.M ^a^ against AChE and BuChE			
		Aerial Part	Root	Flower	Fruit
MeOH	AChE	6.45 ± 2.33	9.58 ± 2.55	^c^	^b^
	BuChE	14.44 ± 1.56	38.12 ± 4.05	27.33 ± 2.65	67.35 ± 1.56
Hexane	AChE	^b^	3.25 ± 1.57	^c^	^c^
	BuChE	17.35 ± 3.08	45.09 ± 2.66	24.97 ± 4.09	34.31 ± 2.76
CH_2_Cl_2_	AChE	^b^	29.15 ± 2.45	^c^	31.46 ± 2.78
	BuChE	64.66 ± 2.56	71.32 ± 3.09	69.25 ± 4.10	82.27 ± 1.97
EtOAc	AChE	3.34 ± 1.49	4.58 ± 4.66	^b^	9.03 ± 2.78
	BuChE	29.09 ± 2.66	28.05 ± 2.13	36.21 ± 2.35	43.44 ± 3.15
BuOH	AChE	3.55 ± 3.70	^c^	4.33 ± 1.65	^c^
	BuChE	17.56 ± 2.54	14.54 ± 3.44	28.23 ± 2.54	^b^
Aqueous residue	AChE	^c^	^b^	^c^	^b^
	BuChE	^b^	9.42 ± 1.97	^b^	17.21 ± 2.45
Essential oils	AChE	^b^	16.66 ± 3.21	6.45 ± 2.09	26.11 ± 2.13
	BuChE	34.56 ± 2.47	56. 30 ± 3.51	78.65 ± 2.66	72.24 ± 2.44
Bergapten	AChE	18.98 ± 2.98			
	BuChE	31.00 ± 3.02			
Imperatorin	AChE	20.44 ± 2.24			
	BuChE	44.23 ± 2.09			
Pimpinellin	AChE	23.54 ± 1.29			
	BuChE	66.55 ± 2.61			
Umbelliferone	AChE	61.09 ± 4.46			
	BuChE	40.99 ± 5.61			
Donepezil	AChE	82.45 ± 2.64			
	BuChE	90.33 ± 4.16			

^a^ Standard error mean, ^b^ No activity, ^c^ Not detected because of turbidity in the wells of microplates.

**Table 3 molecules-24-00722-t003:** *Zosima absinthifolia* Essential oil yields (*w*/*v*, %).

Used Parts	Crushed (g)	Yields	Colour	Collection Time
Aerial	152	0.329	Light yellow	2017
Root	132	0.008	White	2017
Flower	35	0.057	Yellow	2018
Fruit	80	1.250	Yellow	2017

**Table 4 molecules-24-00722-t004:** The composition of the essential oils of *Zosima absinthifolia*.

RRI	Compound	Ap %	R %	Fl %	Fr %
1032	α-Pinene	4.4	1.3	2.2	0.1
1048	2-Methyl-3-buten-2-ol	-	-	tr	tr
1076	Camphene	0.2	-	0.1	tr
1093	Hexanal	0.3	-	tr	-
1118	β-Pinene	2.0	8.9	0.2	0.1
1132	Sabinene	0.3	0.1	0.1	tr
1151	δ-4-Carene	-	-	0.1	-
1174	Myrcene	1.0	3.0	1.3	tr
1176	α-Phellandrene	0.2	0.1	-	-
1194	Heptanal	-	0.4	-	-
1203	Limonene	1.8	2.7	1.5	0.1
1218	β-Phellandrene	1.0	0.4	0.7	0.1
1225	(*Z*)-3-Hexenal	-	-	tr	-
1244	2-Pentyl furan	-	0.2	0.1	tr
1246	(*Z*)-β-Ocimene	-	0.3	-	tr
1255	γ-Terpinene	-	0.2	tr	-
1266	(*E*)-β-Ocimene	-	-	0.4	-
1280	*p*-Cymene	0.5	2.2	0.1	-
1290	Terpinolene	0.4	1.1	0.1	tr
1296	Octanal	0.3	2.5	tr	0.2
1348	6-Methyl-5-hepten-2-one	-	-	tr	-
1360	Hexanol	-	-	tr	-
1398	2-Nonanone	-	2.6		-
1399	Methyl octanoate	-	-	tr	-
1400	Nonanal	-	0.3	tr	-
1444	Ethyl octanoate	-	-	0.2	-
1452	α,*p*-Dimethylstyrene	-	0.3	-	-
1483	Octyl acetate	7.3	1.0	19.9	81.6
1497	α-Copaene	-	-	0.1	tr
1506	Decanal	-	-	-	0.1
1516	(*Z*)-4-Octenyl acetate	0.3	-	0.5	5.1
1535	β-Bourbonene	1.3	-	0.1	0.3
1538	*trans*-Chrysanthenyl acetate	-	-	1.6	-
1553	Linalool	-	-	0.4	0.2
1562	Octanol	8.8	2.8	4.6	3.2
1571	*trans-p*-Menth-2-en-1-ol	-	-	1.5	0.1
1586	Pinocarvone	-	0.5	-	-
1589	β-Ylangene	-	-	-	tr
1591	Bornyl acetate	0.9	0.3	1.3	0.2
1597	β-Copaene	-	-	-	0.1
1600	β-Elemene	-	-	-	tr
1610	Calarene (=*β-*gurjunene)	-	0.2	-	-
1612	β-Caryophyllene	1.8	0.2	1.0	0.2
1614	Carvacrol methyl ether (= methyl carvacrol)	-	0.5	-	-
1623	Octyl butyrate	0.4	-	0.2	0.2
1634	Octyl 2-methyl butyrate	0.5	-	0.4	0.1
1638	*cis-p*-Menth-2-en-1-ol	-	-	0.7	0.1
1648	Myrtenal	-	0.4	-	-
1655	(*E*)-2-Decenal	-	1.5	-	-
1660	(*Z*)-4-Octenyl butyrate	-	-	0.2	-
1661	*trans*-Pinocarvyl acetate	-	26.7	-	0.1
1668	Citronellyl acetate	-	-	1.4	0.1
1670	*trans*-Pinocarveol	-	1.4	-	-
1687	Decyl acetate	-	-	-	0.1
1687	α-Humulene	-	-	0.1	-
1689	*trans*-Piperitol	-	-	0.4	-
1690	Cryptone	-	-	0.2	-
1704	Myrtenyl acetate	-	0.9	-	-
1706	α-Terpineol	-	0.3	-	-
1719	Borneol	-	-	0.1	-
1726	Germacrene D	2.3	-	2.0	0.5
1733	Neryl acetate	-	-	0.1	-
1747	*trans*-Carvyl acetate	-	0.2	-	-
1755	Bicyclogermacrene	0.7	-	0.7	0.1
1758	*cis*-Piperitol	-	-	0.5	-
1758	(*E,E*)-α-Farnesene	-	-	0.2	-
1772	Citronellol	-	-	0.4	0.1
1773	δ-Cadinene	-	-	0.1	-
1779	(*E,Z*)-2,4-Decadienal	-	0.3	-	-
1786	*ar*-Curcumene	0.2	0.3	0.2	0.1
1689	*trans*-Piperitol	-	-	-	tr
1804	Myrtenol	-	0.6	-	-
1827	(*E,E*)-2,4-Decadienal	-	0.9	-	-
1829	Octyl hexanoate	0.7	-	0.6	0.2
1849	Cuparene	-	0.6	0.1	-
1856	(Z)-4-octenyl hexanoate	0.7	-	0.7	-
1857	Geraniol	-	-	0.2	0.1
1868	(*E*)-Geranyl acetone	-	1.4	0.1	-
1878	2,5-Dimethoxy-*p*-cymene	-	3.6	tr	-
1945	1,5-Epoxysalvial(4)14-ene	-	-	tr	-
1958	(*E*)-β-Ionone	-	-	0.3	-
1981	Heptanoic acid	-	0.2	-	-
2000	Citronellyl hexanoate	-	-	0.3	-
2008	Caryophyllene oxide	1.9	0.8	0.4	0.1
2020	Octyl octanoate	7.6	0.8	0.3	0.9
2050	(*E*)-Nerolidol	0.8	-	0.1	-
2069	Germacrene D-4β-ol	-	-	0.3	-
2084	Octanoic acid	-	-	-	0.1
2100	Heneicosane	-	-	0.1	-
2127	10-*epi*-γ-Eudesmol	-	-	0.1	-
2131	Hexahydrofarnesyl acetone	0.4	-	0.1	tr
2144	Spathulenol	2.7	-	0.5	0.1
2170	β-Bisabolol	-	-	0.1	-
2183	γ-Decalactone	-	-	-	0.1
2187	T-Cadinol	-	-	0.1	-
2192	Nonanoic acid	-	-	-	0.1
2200	Docosane	-	-	0.1	-
2209	T-Muurolol	-	-	0.2	-
2214	(2E,6Z)-Farnesal	-	-	0.1	-
2219	δ-Cadinol (= torreyol)	-	-	0.1	-
2247	*trans*-α-Bergamotol	-	-	0.1	-
2255	α-Cadinol	-	-	0.6	-
2271	(2*E*,6*E*)-Farnesyl acetate	-	-	2.3	-
2278	(2E,6E)-Farnesal	-	-	0.4	-
2300	Tricosane	-	-	0.7	-
2373	Unknown I	12.5	5.0	15.4	1.0
2369	(2*E*,6*E*)-Farnesol	-	-	1.7	-
2450	Unknown II	26.4	2.3	8.9	1.4
2500	Pentacosane	-	-	-	0.1
2503	Dodecanoic acid	-	-	-	0.2
2622	Phytol	-	-	0.5	-
2670	Tetradecanoic acid	-	-	-	1.0
2700	Heptacosane	-	-	0.5	-
2900	Nonacosane	-	-	-	0.2
2931	Hexadecanoic acid	4.1	1.3	0.5	0.4
	**Total Identified**	55.8	74.3	58.2	96.8
	***Total***	*94.7*	*81.6*	*82.5*	*99.2*

RRI: Relative retention indices calculated against n-alkanes; % calculated from FID data; tr Trace (< 0.1 %); Ap: Aerial part; R: Root; Fl: Flower; Fr: Fruit; Unknown I: EIMS, 70 eV, m/z (rel. .int.): 270[M]^+^(0.4), 227(41), 159(6), 141(37), 115(97), 98(100), 81(33), 69(23), 57(19), 43(66); Unknown II: EIMS, 70 eV, m/z (rel. .int.): 228[M]^+^(0.5), 210(0.5), 116(25), 98(100), 87(19), 71(23), 57(18), 41(26).

**Table 5 molecules-24-00722-t005:** Chemical class distribution of the samples.

Compound Class	Ap %	R %	Fl %	Fr %
Esters	9.4	29.1	27.9	87.5
Alcohols	8.8	5.1	8.8	3.8
Aldehydes	0.6	6.3	0.5	0.3
Ketones	0.4	4.5	0.7	tr
Fatty acids+ esters	13.1	2.3	2.6	2.9
Terpene hydrocarbons	18.1	21.9	11.4	1.7
Oxygenated terpenes	5.4	4.9	4.8	0.2
Furans	-	0.2	0.1	tr
Alkanes	-	-	1.4	0.3
Lactones	-	-	-	0.1
**Total Identified**	55.8	74.3	58.2	96.8

**Table 6 molecules-24-00722-t006:** Weights of the crushed plants and obtained extracts and fractions.

Species	Extracts/Fractions	Aerial Part	Root	Flower	Fruit
*Zosima absinthifolia*	MeOH (g)	25.01	29.88	23.92	85.98
	Hexane (g)	3.28	4.05	2.98	11.88
	CH_2_Cl_2_ (g)	9.12	10.10	8.97	26.01
	EtOAc (g)	1.66	2.24	1.59	4.81
	BuOH (g)	4.92	5.86	4.77	18.57
	Aqueous residue (g)	5.02	3.22	4.98	6.96

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
