# Peer review of "Molecular Docking Studies of Coumarins Isolated from Extracts and Essential Oils of Zosima absinthifolia Link as Potential Inhibitors for Alzheimer’s Disease"

_molecules, 2019, doi:10.3390/molecules24040722_

Round 1

Reviewer 1 Report

Interesting and scientifically solid work. The manuscript merits publication but after extensive editing of English language.

Author Response

Thank you Reviewer for the comment. We corrected English text to improve it. 

Reviewer 2 Report

The article deals with the investigation of antioxidant and anticholinesterase activity of extracts and essential oils from Zosima absinthifolia, aimed at inhibiting Alzheimer's disease.

First, in my opinion, the quality of language and grammar alone would justify plain rejection. It looks like the Authors didn't even bother to use a built-in word corrector available in any word processor. This is really a pity, because, in principle, the study would be quite interesting, and the latter is the reason why I have spent a lot of time and effort to try suggesting radical improvements to the manuscript, and suggest only major revision.

More substantially:

The Title is too long and should be simplified (see the herein enclosed document).

Motivation of the study is not clearly stated. For example, insufficient action and/or unfeasibility of other phytochemicals / natural products, as well as, if applicable, possible superiority of the investigated substances.

Indeed, it is hard to understand from the text whether the tested plant materials are more or less effective than others. Some comparisons with literature results from other plant materials should be introduced.

In relative terms, the Introduction is not so bad, but the state of the art is insufficiently depicted.

Overall, the research design is reasonably well-made, but some parts (for example, about secretory canals) are disconnected and hard to understand in the context.

The study results are poorly presented, and some statements are unclear, unsupported, or seemingly wrong.

The Section "Results and discussion" is too long and should be splitted in two separate Sections.

The text about the secretory canals is disconnected from the rest of the manuscript, i.e., its meaning and consequences are unclear.

Several statements in the Section "Materials and Methods" are confused, or poorly documented and uniformative.

 The Conclusions section should be deeply reworked and streamlined, in order to convey the most important messages drawn from this study. In the present form, Conclusions consist of a series of stand-alone, poorly coordinated statements.

Due to the great deal of comments and corrections (about 140), all my comments and corrections can be found in the herein enclosed document.

The Authors are warmly invited to read carefully not only the above comments, but also all the comments and corrections in the herein enclosed document, and react accordingly, before submitting the revised version of the manuscript. Only after such heavy revision effort, the manuscript could be considered for publication.

Author Response

Dear Reviewer,

Thank you very much for the great work which you done. We followed all your comments which really were helpful to improve text. Please, see the corrected parts were coloured with yellow.

The next changes were done:

-          The language of manuscript was edited according to the app.grammarly.com.

-          The Title was revised.

-          Coumarins and essential oils are the major components of the Apiaceae family and Zosima genus, so we focused tha that compounds and isolated them.

-          Many matters could be grouped as irregular AD, where a major risk factor is age was rearranged.

-          The sentence “it is rather substantial to 58 own both anticholinesterase and antioxidant potentials for a drug nominee towards AD” was revised.

-          The reference 6 was changed.

-          ROS was rewritten.

-          Antioxidants may clean ROS and break inflammatory pathways. The utilization of antioxidants is useful in AD progress for this sentence “8.    Albanese, L.; Ciriminna R.; Meneguzzo F.; Pagliaro, M. Innovative beer-brewing of typical, old and healthy wheat varieties toboost their spreading. Journal of Cleaner Production 2018, 171, 297–311.” was added.

-          Antioxidants may clean ROS and break inflammatory pathways. The utilization of antioxidants is useful in AD progress [7]. was revised.

-          Results and discussions were divided two parts.

-          The figure 2 was improves qulaity. The explanation for antioxidant activity was rewritten with next point that DPPH analysis results were showed the presence of antioxidant activity in the range from 61.92- 69.2 % which can is not really high. The values for the content of total phenolics content in the root and fruit part were added in the text.

-          “the antioxidant activity results of extract from different Z. absinthifolia parts were higher compared the standards propyl gallate, chlorogenic acid, and rutin” this sentence was revised.

-          TBA was explained.

-          IC50 values were explained at material and method part.

-          “Many of samples indicated antioxidant activity on liposome in comparison to the rutin and chlorogenic acid.” this sentence was revised.

-          “We suppose that it can be related to presence another secondary metabolite which are liable for the antioxidant activity.” This sentence was removed.

-          20 mg/mL was revised.

-          The Figure 4 was rechanged for pimpinellin.

-          Molecular study is related to the pharmacetical chemistry and is a different region, so the results are so clear and understandable.

-          “The colour of essential oils from different parts of Z. absinthfolia were varied.”this part is necessary, because the colours of essential oils mean that theie chemical components are different.

-          At Table 5 furans were revised.

-          The text “Nowadays is not presented information about the presence of phenolic compounds in the Zosima genus, especially such as α-pinene [31], octyl acetate [32], and which characterized by antioxidant potential. The high abundance of octyl acetate and possible antioxidative and anticancer effects were estimated in essential oils from the leaves of species of Pittosporum (Pittosporaceae) [33] and in essential oils from Ethiopian herbs Boswellia carterii and Commiphora pyracanthoides Engler [34]was rewritten to make text easy to read and underanstanble. Now it is look like that: Nowadays is not presented data about the presence of phenolic compounds in the essential oil from Zosima genus, especially such as α-pinene and octyl acetate which in previous studies shown high antioxidant capacity [33, 34].”

-          Thank you Reviewer for the right comment regarding mistake in the DPPH method. It was rewritten sentence about blank sample to make method description clear.  Please, see changes in the yellow color: “A sample without DPPH solution was utilized as a blank sample.”

-          “manually cross sections from the peduncles, rays, pedicels, and fruits samples for Z.absinthifolia” was rewritten.

-          “Most active compounds (against AChE: Umbelliferon, against BuChE: Pimpinellin) were 391 docked at the binding sites of 1-EVE and 1-P0I.” was cleared.

-          “Finally the 398 resstrained minimization the added hydrogen atoms was performed with OPLS3e.” was cleared.

-          Statistical analysis part was rewritten.

-          RSC was written as radical scavenging capacity.

-          The Conclusions section was revised.

Reviewer 3 Report

The subject of the research is very interesting and the results are novel and valuable. The technical part of the study is generally well made, however, some parts of description of applied methods are missing. I understand that the detailed description can be omitted when it is available in the reference. However, it would be better at least to mention the main name of the classical method, for example for determination of anti-lipid peroxidation the paper Karakaya, S.; Koca, M.; Kilic, C.S.; Coskun, M. Antioxidant and anticholinesterase  activities (…) is cited, but it would be suitable to mention that it was TBA method (as DPPH method was mentioned for antioxidant properties).

However, I would strongly suggest to improve English language used for writing this manuscript. Generally it is understandable (although awkward) but there are numerous grammar mistakes, or letters missing etc.

Some fragments of the text can therefore be hard to understand for the reader, e.g., MeOH extracts were fragmented utilising solvers withal varied polarities. I guess it means that MeOH extracts were fractionated with the use of different solvents?

The sentence “A sample with no supplemented samples was utilized as a blank sample”(line 368) is a real curiosity…

I consider as really awkward the style of the sentence: “The findings of samples regarding content of total phenolics are presented in Figure…”, “The dichloromethane fraction of fruit had the best inhibition towards butyrylcholinesterase” or “Secretory canals of the fruits and pollens of Zosima species were as described in advance”  etc.

These examples are too numerous to correct them in the review, I think that the manuscipt requires proofreading before publication.

Author Response

Dear Reviewer,

Thank you very much for the useful comments, especially connected to the part "Material and methods". We are fully agree with comments which are really were able to improve body and main idea of MS.

The English correction regarding grammar mistakes, or letters missing etc. were done and text was revised.

The next responces regarding changes which were done in the text you can see bellow. The changes in the revised version of MS are presented in the yellow color.

-          Anti-lipid peroxidation activity was written as DPPH method.

-          Thank you, Reviewer, for the right comment regarding mistake in the DPPH method. It was rewritten sentence about blank sample to make method description clear.  Please, see changes in the yellow color: “A sample without DPPH solution was utilized as a blank sample.”

-          MeOH extracts were fragmented utilising solvers withal varied polarities. Was changed as MeOH extracts were fractionated with the use of different solvents.

-          The findings of samples regarding content of total phenolics are presented in Figure…”, “The dichloromethane fraction of fruit had the best inhibition towards butyrylcholinesterase” or “Secretory canals of the fruits and pollens of Zosima species were as described in advance”  etc. were rewritten.

-          The reference (Kim, J.; Lee, H.J.; Lee, K.W. Naturally occurring phytochemicals for the prevention of Alzheimer's disease. Journal of Neurochemıstry 2010, 112, 1415–1430. [CrossRef]) the reviewer suggested was added.

-          The reference (Albanese, L.; Ciriminna R.; Meneguzzo F.; Pagliaro, M. Innovative beer-brewing of typical, old and healthy wheat varieties toboost their spreading. Journal of Cleaner Production 2018, 171, 297–311. [CrossRef]) the reviewer suggested was added.

-          The reference (Ciriminna, R.; Meneguzzo, F.; Fidalgo, A.; Ilharco, L.M.; Pagliaro, M. Extraction, benefits and valorization of olive polyphenols. Eur. J. Lipid Sci. Technol. 2016, 118, 503–511. [CrossRef]) the reviewer suggested was added.

Round 2

Reviewer 2 Report

The authors managed to revise the manuscript quite accurately, complying with my comments as well as the comments produced by the other esteemed Reviewers.

Now, the text has become attractive, streamlined, and definitely very well readable. As well, all statements are properly supported.

Only few residual (or newly introduced) language errors remain, which are listed below:

Line 61. "improving" --> "improvements".

Line 63. Remove "recent".

Line 74. "A lot of investiagtion were done" --> Much investigation was performed.

Line 85. "at food" --> "in food"; "Farther" --> "Further".

Line 90. "are the largest predicament" --> "comprise most".

Line 91. "augments" --> "is growing".

Lines 93-94. "in each of the three parts(" --> "for the purpose of"; ") including" --> ", as well as".

Line 144. "considerable high compared" --> "considerably high compared with".

Line 285. "high" --> "higher".

Line 286. "comparison to" --> "in comparison with".

Line 299. "Nowadays is not presented data" --> "So far, no data are available".

Line 301. "shown" --> "showed".

Lines 318-320. Remove paragraph because it is useless.

Line 326. "withal" --> "with".

Line 337. "role" --> "act".

Once the authors will have fixed these minor mistakes, the manuscript can be published.

Author Response

Dear Reviewer,

Thank you very much for the comments and corrections!

Please, see changes in the text in yellow color and answers to the your comments bellow.

RESPONSE TO THE REVIEWERS

-          The corrected parts were coloured with yellow.

-          Line 61. "improving" was changed as "improvements".

-          Line 63. Remove "recent". was removed.

-          Line 74. "A lot of investiagtion were done" was changed as Much investigation was performed.

-          Line 85. "at food" --> "in food"; "Farther" --> "Further". Were revised.

-          Line 90. "are the largest predicament" was changed as "comprise most".

-          Line 91. "augments" was changed as "is growing".

-          Lines 93-94. "in each of the three parts(" --> "for the purpose of"; ") including" --> ", as well as".were revised.

-          Line 144. "considerable high compared" was changed as "considerably high compared with".

-          Line 285. "high" was changed as "higher".

-          Line 286. "comparison to" was changed as "in comparison with".

-          Line 299. "Nowadays is not presented data" was changed as "So far, no data are available".

-          Line 301. "shown" was changed as "showed".

-          Previously, secretory canals of the fruits and pollens of Zosima species were studied [39]. The number of secretory canals were the most at peduncle while the shape of secretory canals were the largest at the fruit. The number of secretory canals was the least at pedicel. was removed.

-          Line 326. "withal" was changed as "with".

-          Line 337. "role" was changed as "act".